# Chronological Trends and Mercury Bioaccumulation in an Aquatic Semiarid Ecosystem under a Global Climate Change Scenario in the Northeastern Coast of Brazil

**DOI:** 10.3390/ani11082402

**Published:** 2021-08-13

**Authors:** Fernando Morgado, Ruy M. A. L. Santos, Daniela Sampaio, Luiz Drude de Lacerda, Amadeu M. V. M. Soares, Hugo C. Vieira, Sizenando Abreu

**Affiliations:** 1CESAM—Centre for Environmental and Marine Studies, Department of Biology, Campus de Santiago, University of Aveiro, 3810-193 Aveiro, Portugal; ruysantos@ua.pt (R.M.A.L.S.); d.sampaio@ua.pt (D.S.); asoares@ua.pt (A.M.V.M.S.); hugovieira@ua.pt (H.C.V.); siz@ua.pt (S.A.); 2Laboratório de Biogeoquímica Costeira, Instituto de Ciências do Mar (LABOMAR), Universidade Federal do Ceará, Fortaleza 2853, Brazil; pgcmt@labomar.ufc.br

**Keywords:** mercury chronological trends, annular tree rings, sediment cores, fish mercury bioaccumulation, semiarid tropical estuary, climate change

## Abstract

**Simple Summary:**

Managing aquatic systems is becoming increasingly complex due to human impacts, multiple and competing water needs and climate variability. Considering the Hg concentration present in the top layers of sediment (~20 cm around 30 to 40 years) with the outer layers in the tree cores tree rings cores and in the sediment’s cores from Pacoti estuary and the Ceará estuary, overall data indicate an increase in mercury in recent years. A positive and significant correlation (*p* < 0.05) was revealed between Hg trends in sediments and Hg trends in annular tree rings. This shared Hg pattern reflects local environmental conditions. The results of this work reinforce the indicators previously described in the semiarid NE region of Brazil, showing that global climate change and some anthropogenic factors are key drivers to Hg exposure and biomagnification for wildlife and humans. Possible climate-induced shifts in these aquatic systems highlight the need for accurate and regionally specific metrics of change in the past in response to climate and for improved understanding of response to climate factors. These processes are inducing a greater mobilization of bioavailable Hg, which could allow an acceleration of the biogeochemical transformation of Hg.

**Abstract:**

Due to global warming, in the northeastern semiarid coastal regions of Brazil, regional and global drivers are responsible for decreasing continental runoff and increasing estuarine water residence time, which promotes a greater mobilization of bioavailable mercury (Hg) and allows increasing fluxes and/or bioavailability of this toxic trace element and an acceleration of biogeochemical transformation of Hg. In this work, an application of dendrochemistry analysis (annular tree rings analysis) was developed for the reconstruction of the historical pattern of mercury contamination in a contaminated area, quantifying chronological Hg contamination trends in a tropical semiarid ecosystem (Ceará River Estuary, northeastern coast of Brazil) through registration of mercury concentration on growth rings in specimens of *Rhizophora mangle* L. and using the assessment in sediments as a support for the comparison of profiles of contamination. The comparison with sediments from the same place lends credibility to this type of analysis, as well as the relationship to the historical profile of contamination in the region, when compared with local data about industries and ecological situation of sampling sites. In order to evaluate the consequences of the described increase in Hg bioavailability and bioaccumulation in aquatic biota, and to assess the biological significance of Hg concentrations in sediments to fish and wildlife, muscle and liver from a bioindicator fish species, *S. testudineus*, were also analyzed. The results of this work reinforce the indicators previously described in the semiarid NE region of Brazil, which showed that global climate change and some anthropogenic factors are key drivers of Hg exposure and biomagnification for wildlife and humans. Considering the Hg concentration present in the top layers of sediment (~20 cm around 15 to 20 years) with the outer layers in the tree ring cores and in the sediment’s cores from Pacoti estuary and the Ceará estuary, overall the data indicate an increase in mercury in recent years in the Hg surface sediments, especially associated with the fine sediment fraction, mainly due to the increased capacity of small particles to adsorb Hg. There was revealed a positive and significant correlation (*p* < 0.05) between Hg trends in sediments and Hg trends in annular tree rings. This shared Hg pattern reflects local environmental conditions. The Hg concentration values in *S. testudineus* from both study areas are not restrictive to human consumption, being below the legislated European limit for Hg in foodstuffs. The results from *S. testudineus* muscles analysis suggest a significant and linear increase in Hg burden with increasing fish length, indicating that the specimens are accumulating Hg as they grow. The results from both rivers show an increase in BSAF with fish growth. The [Hg] liver/[Hg] muscles ratio >1, which indicates that the *S. testudineus* from both study areas are experiencing an increase in Hg bioavailability. Possible climate-induced shifts in these aquatic systems processes are inducing a greater mobilization of bioavailable Hg, which could allow an acceleration of the biogeochemical transformation of Hg.

## 1. Introduction

Mercury (Hg) is a neurotoxin and highly toxic, persistent and global pollutant that accumulates in aquatic systems and, due to its persistence in the environment and the possibility of biomagnification through the food web [1,2,3], represents a risk for wild animals and ultimately the human population [1,2,4,5]. Unlike organic contaminants, Hg does not undergo microbial or chemical degradation, persisting in its original quantities after its introduction into the environment, accumulates in living organisms and can cause physiological, neurological, behavioral and reproductive diseases and disorders, harmful to wildlife [2,6]. The high toxicity and the dietary exposure to methylmercury (MeHg) production and its biomagnification through the food chain pose the risk of increased MeHg and adverse Hg-related health effects to top predatory animals, particularly those associated piscivorous species, birds and mammals [6,7,8]. The toxic effects of Hg in fish and wildlife have been extensively reviewed [6,7,8,9,10,11,12,13]. Several physiological and immunologic Hg adverse effects have been described [14,15,16,17,18], as well as effects relating to nephrotoxicity [19], population and behavior (neurological and neurobehavioral function) [20,21,22,23,24,25], endocrine disruption [26,27] and reproduction [28,29,30,31,32]. Currently, fish and wildlife are much less exposed to acute high dietary MeHg than to chronic exposure scenarios [33]. Nevertheless, tissue Hg concentrations in several fish and wildlife species in some environments are within an order of magnitude of levels associated with overt toxicity [20,34] and may be sufficiently high to cause reproductive impairment, behavioral abnormalities or other, subtler biological effects [28]. Due to the diverse environmental sources, the various exposure pathways through food webs and from industrial activities, its volatility and accompanying long-range atmospheric transport, its toxicity and its ability to bioaccumulate in organisms [1,4], it is crucial to characterize the adverse health effects of Hg and manage the risk of Hg exposure to wildlife and public health. [6,7,9,13,35]. These characteristics make Hg and its compounds priority pollutants, under European Union regulations (EU, number 231-106-7), the Water Framework Directive, the OSPAR Convention, the United Nations Environment Programm and environmental agencies across the world [36].

The extent and intensity of some drivers of climate change, land use changes and Hg emissions will continue to threaten key processes of wild organisms and humans, and should be seen as a global problem, since Hg can be transported over long distances by atmospheric processes [1,3,37,38]. Considering atmospheric deposition, it is widely accepted that a cold and dry climate is conducive to Hg accumulation, while a warm and humid climate limits the accumulation of Hg [39,40,41,42,43]. Although, orographic precipitation can also enhance Hg deposition in humid climate periods [44]. Estimates of anthropogenic emissions show that they currently contribute about two-thirds of the current global Hg cycle, raising the global concentration to four to five times that of the background levels [45,46,47]. These widespread environmental and health questions have earned international recognition given the relevance of understanding Hg remobilization processes and their response to global environmental changes.

The effects of climate change on increasing Hg contamination have been particularly enhanced in ecosystems exposed to climatic extremes, mainly in the transition (estuarine) ecosystems [1,3,41,48]. Continental runoff in semiarid regions is affected by land use and global climate change, leading to the increasing accumulation of particulate Hg in estuaries and increasing export of dissolved Hg to the ocean [49]. This scenario is especially relevant in the food webs of tropical aquatic ecosystems, especially in the estuaries and continental shelf of the northeastern regions of Brazil, being already subject to semiarid conditions, but drought events are now intensifying in terms of frequency and duration of episodes. Recent studies developed in this region of the world, mainly in the Ceará estuary, reported that these episodes have increased from 1 to 3 per century during the 16th and 17th centuries to more than 14 in the 20th century [50]. Formerly, this diminution was caused by river damming for water consumption due to a growing urban population and for irrigated agriculture, followed by land use change; in particular, conversion of gallery forests to pasture and agriculture [51]. In the semiarid region of NE Brazil, similarly to that recently verified in the Artic [52,53,54,55], the regional and global drivers responsible for increasing continental runoff and diminishing fluvial discharges to the Western Equatorial South Atlantic Ocean are strengthening due to global warming [56]. Lacerda et al. [49] verified in 2009 that during the rainy season, a positive net flux to the ocean occurred but, even during the extreme rainy period, marine waters contributed significantly to the total flow, mostly from the estuary to the sea, with a much larger flow compared to flow from the fluvial to the estuary. The same study reported that during the dry season, the discharge to the ocean was mostly due to tidal waters entering the estuary, with a small contribution from the higher basin. Climate change could be the most important reason for the intensification of this process [56]. At different geographic scales, the interaction of these factors establishes chronological profiles with different temporal variations of contamination specific to each ecosystem [5,57,58]. Therefore, it is crucial to analyze climate change’s effects on the intensification of some mechanisms triggering higher Hg bioaccumulation in the aquatic ecosystems of this region. Seawater intrusion will promote not only higher resuspension processes, but also longer water residence time in the dry season, favoring intra-estuary reactions of continental and marine-born materials brought in by fluvial and tidal flows (Lacerda et al., 2013), thus leading to an increase in bioaccumulation.

Accumulation trends have been inferred mostly from records in sediment layers, but in recent years, dendrochemistry has provided great insights into environmental and social histories. Biologists, climatologists and physical geographers have used dendrochronological techniques to investigate environmental changes and recent global climate changes in different parts of the world [37,38,58,59,60,61,62,63]. Dendrochronological methods have successfully dated and interpreted geological and hydrological conditions [64], avalanches and rock falls [65], archaeological sites’ human activity [4] and climate change [66,67,68,69,70,71]. Tree rings have already been used as biogeochemical tracers to map the extent of environmental pollution due to anthropogenic emissions by different countries [37,62,72]. Several studies have also used dendrochemistry to address environmental pollution from different sources [37,58,62,63,72]. However, only a few studies have directly linked the levels of metal traces in wood to changes in emissions from suspected sources of pollution [37,62,63,72]. Annular tree rings have been suggested as useful tools to map the biogeochemical extent of environmental Hg pollution, assuming that the growth ring of a given year partially reflects the chemicals available in the environment and events and trends in ecosystem processes, on a spatial and temporal scale, over decades [73]. The basic assumption in tree ring chemistry (dendrochemistry) is the stability of the element distribution associated with no significant mobility after storage [74].

This approach may be an important contribution to clarify the global climate change increase in Hg bioavailability and bioaccumulation in aquatic biota of tropical, semiarid ecosystems, as a monitory system of the effects on wildlife and biota. The lack of records on the baseline conditions for the period prior to the start of exposure in a given ecosystem has been a major gap in understanding the behavior and historical evolution of a contaminant over time. The knowledge of the behavioral pattern of a contaminant in an ecosystem allows evaluating and integrating into this profile the most recent levels of contamination, inferring about a possible regression or progression of the levels of contamination observed, either on a local or global scale [5,7]. Thus, it is possible to hypothesize that growth rings can provide the biological recording matrix for the chronological exposure profile, assuming that when a ring loses functions, the tissue retains and isolates the chemical components of the sap corresponding to that date. The sequence of records stored in the various rings along the trunk growth of a tree would provide the chronological evolution of exposure to the contaminant [37,62,72]. This paper intends to contribute to a better understanding of the continued fulfillment concerning the global cycle of Hg in this region, under a global climate change scenario, analyzing, in the framework of the new theory “Arctic Paradox” [56], the increase in Hg bioavailability and bioaccumulation in aquatic biota of tropical, semiarid ecosystem (Ceará River Estuary, northeastern coast of Brazil). We analyzed the registration of Hg concentration on growth rings in specimens of *Rhizophora mangle* L, and used the assessment in sediments as a support for the comparison of profiles of contamination. In order to evaluate the consequences of the described increase in Hg bioavailability and bioaccumulation in aquatic biota and to assess the biological significance of Hg concentrations in sediments to fish and wildlife, muscle and liver from a bioindicator fish species, *S. testudineus*, were also analyzed. The hypothesis was that with increasing residence time of water in the estuary, there is a greater mobilization of bioavailable Hg, which allows an acceleration of the biogeochemical transformation of Hg.

## 2. Materials and Methods

### 2.1. Study Area

The study area was located in the Metropolitan Region of Fortaleza, capital of Ceará State, northeastern coast of Brazil, using a contaminated area (Ceará River estuary) and a reference site (Pacoti River estuary). The Pacoti River rises in the hills of Baturité with a course of approximately 130 Km to the southwest–northeast [75] (Figure 1). The river basin is marked by the presence of dams. Most of the estuary is inserted into the “APA”—an Environmental Protection Area, which, since 2000, has tried to reduce the degradation of local mangroves [76]. This protected area was initially viewed as a reference area for comparison of data in relation to contamination and its effects on organisms because it is a region with lower rates of urbanization and industrialization [77]. However, a compromise was observed in the quality of the estuary, classifying the region as “compromised”, but to a lesser degree of degradation when compared with other estuaries in the Metropolitan Region of Fortaleza [77]. Due to ecological and economic importance, the Ceará River is subjected to degradation resulting from the high industrial and urban development [78].

### 2.2. Study Sampling Design

The study sampling design included an application of dendrochemistry analysis (annular tree rings analysis) to the reconstruction of the historical pattern of Hg contamination, through registration of Hg concentration on growth rings in specimens of *Rhizophora mangle* L. and using the assessment in sediments as a support for the comparison of profiles of contamination. The comparison with sediments from the same places lends credibility to this type of analysis as well as the relationship to the historical profile of contamination in the region, when compared with local data about industries and the ecological situation of sampling sites (Figure 2). In order to evaluate the consequences of the described increase in Hg bioavailability and bioaccumulation in aquatic biota, and to assess the biological significance of Hg concentrations in sediments to fish and wildlife, muscle and liver from a bioindicator fish species, *S. testudineus*, were also analyzed.

### 2.3. Rhizophora mangle L. (Red Mangrove) Sampling and Treatment

Specimens of *Rhizophora mangle* L. (Red Mangrove) were chosen by the rapid response of mangroves to the regional and global environmental changes [77]. Ceará estuary margins: Ceará A (3°43′11.06″ S; 38°37′22.4″ W), Ceará B (3°42′59.1″ S; 38°37′16.75″ W) and Ceará C (3°42′73″ S; 38°36′52.73″ W); and point of Pacoti estuary (3°49′52.2″ S; 38°25′11.78″ W) as reference (Figure 1). Cores (three trees in each site) were taken at breast height (*1.5 m) from each tree using a stainless-steel borer (5 mm in diameter and 300 mm in length) (Figure 2). The trees were about 10 cm in radius. The cores were removed at only 5 to 6 cm in depth on the side of the trunk and not on the entire trunk (10 cm), because a greater depth could cause distortion and compression of the core. After sampling, the cores were stored keeping their original shape and preventing degradation. In the laboratory, the cores were divided into 1 cm pieces and stored individually in Eppendorf tubes, properly identified. Trees were planted in the 1980s, meaning that the older trees were about 30 years old and showed about 10 cm radial growth from the tree center; thus, each centimeter radial segment corresponds to approximately 3 years of tree growth. The cores were cut, stored individually in Petri dishes and placed in an oven at 30° for 3 days.

### 2.4. Sphoeroides Testudineus Sampling and Treatment

*S. testudineus* (also known as checkered puffer) was chosen in the present work for several reasons: (1) its high abundance in tropical nvironments; (2) it had already proven to be very useful as a Hg bioindicator [79]; (3) in juvenile stage, the species tends to orient towards the substrate for feeding and defense, being in close contact with the contaminated sediment particles; and (4) they are intermediate predators in the estuarine food chain [80,81]. Juvenile *S. testudineus* (n = 15) were collected in Ceará River estuary (3°42′ S; 38°36′ W) and in a reference site, Pacoti River estuary (3°49′ S; 38°25′ W), using a hand-operated trawl net during low tide. The fishes were transported to the laboratory in oxygenated local water buckets immediately after capture. Muscle and liver samples were selected from each fish and stored frozen at −20°C until being lyophilized and analyzed [82,83].

### 2.5. Surface Sediment and Water Sampling and Treatment

In order to assess present Hg contamination in Ceará River estuary, 3 replicates of water and surface sediments were sampled during low tide in three stations: Ceará A (3°43′11.06″ S; 38°37′22.4″ W), Ceará B (3°42′59.1″ S; 38°37′16.75″ W) and Ceará C (3°42′73″ S; 38°36′52.73″ W) and in one point of Pacoti estuary (3°49′52.2″ S; 38°25′11.78″ W) as a reference (Figure 1). Surface sediment (i.e., 0–2 cm removable bed particles) cores of about 20 cm were collected as composite samples and transported to the laboratory in carefully labeled plastic bags, where they were oven dried at 40 °C for 3 days, and later homogenized and sieved continuously with stainless steel sieves with mesh sizes of 2 mm, 1 mm and 63 μm.

The sediment cores were cut into 1 cm thick slices. A first Hg quantification was made every 5 cm. In places where it seemed to indicate some oscillation/variation, more quantifications were made (2 cm and 2 cm). That is why some colors have “fine” quantification and others only every 5 cm (coarse sediment). Sediment fractions between 1 mm and 63 μm (coarser sediment) and <63 μm (fine sediment) were used for Hg quantification (Figure 2). Organic matter content in sediments was estimated as weight loss on ignition (LOI). In parallel, water samples were also collected in the same stations using 5 L polyethylene bottles, and immediately filtered in the laboratory using polyethylene filter holders (previously acid cleaned) and weighted millipore filters of 0.45 µm in order to separate the suspended particulate matter (SPM) and the dissolved matter fraction. After the filtration process, filters were oven dried at 40 °C overnight and stored until Hg analysis.

### 2.6. Mercury Analysis and Data Quality Control

Mercury quantification was performed by atomic absorption spectrophotometry after thermal decomposition of the sample (AMA 254 Trace Mercury Analyzer, LECO, St. Joseph, MI, USA). Mercury concentrations are expressed as µg g^−1^ on a dry weight basis. To check for contamination, blanks were also analyzed using this procedure after every 5 samples. Quality control was evaluated using Certified Reference Materials (TORT-2 and PACS-2). To evaluate the accuracy and precision of the analytical methodology, reference fish material TORT-2 and Marine Sediment Reference Material PACS-2 were analyzed in parallel with all samples. Experimental values of 0.26 ± 0.01 µg g^−1^ for TORT-2 and 3.02 ± 0.08 µg g^−1^ for PACS-2 were within the confidence level (0.27 ± 0.06 µg g^−1^ and 3.04 ± 0.2 µg g^−1^, respectively).

### 2.7. Biota–Sediment Accumulation Factor Determination

Biota–sediment mercury contamination interaction was assessed comparing Hg accumulation in *S. testudineus* fish muscle and Hg concentration values found on a fine fraction of surface sediments, and was calculated following the formula suggested by Green-Ruiz et al. [84]:(1)$$\mathrm{BSAF}=\frac{\left[\mathrm{Mercury}\right]\;\mathrm{in}\;\mathrm{organism}}{\left[\mathrm{Mercury}\right]\;\mathrm{in}\;\mathrm{associated}\;\mathrm{sediment}}$$

BSAF was used to classify the aquatic species as macroconcentrator (BSAF > 2), microconcentrator (1 < BSAF < 2) or deconcentrator (BSAF < 1) [85].

### 2.8. Geoaccumulation Index (Igeo)

The degree of Hg pollution was calculated using the geoaccumulation index originally defined by Muller [86]. *Igeo* can be calculated by using the following formula:(2)$$Igeo=log_{2}\left(\frac{Cn}{1.5\;\left(Bn\right)}\right)$$

Here, *Cn* is the measured concentration of the metal in the sediment, *Bn* is the background value of the metal value and 1.5 is the factor used to account for possible variations in the background caused by lithological variations. The geoaccumulation index consists of seven grades or classes (Table 1).

### 2.9. Statistical Analyses

Differences between sampling site sediments were tested using one-way analysis of variance (ANOVA), followed by the Dunnett’s comparison test whenever applicable. Departures from normality in Hg concentration data from the 63–1000 μm fraction were corrected with √(x) transformation and differences were considered statistically significant when *p* < 0.05.

## 3. Results

### 3.1. Mercury in Tree and Sediment Cores

Considering the Hg concentration present in the tree ring cores, a lower Hg concentration was observed in the Pacoti estuary (ranging from 0.6 to 1.7 ng g^−1^) when compared to the Ceará estuary. In this estuary, Hg concentration ranged from 1.1 to 31 ng g^−1^ at site A, between 0.78 and 12 ng g^−1^ at site B and 2.0 to 11 ng g^−1^ in site C. At site A, the concentration of Hg in the outer rings is 20× higher; however, the inner rings show a decrease in the concentration of Hg (Figure 3). In addition, the sediment cores from the Pacoti estuary also presented relatively high levels of Hg (ranging from 10 to 20 ng g^−1^) when compared to the Ceará estuary. The highest levels of Hg were observed mainly in the upper layers of the Ceará A station, where the concentration of Hg was much higher, but decreasing to levels similar to the reference site (after 30 cm) (Figure 3).

Comparing Hg levels downstream of Ceará estuary, from Ceará A to Ceará C, the results indicate a decreasing Hg contamination, registered both in sediment and tree ring cores. Station A is the most contaminated site, and also presents a historical Hg pattern of accumulation, which is similar both in sediment and annular tree ring cores. The Hg accumulation evolution pattern in the last 15 to 20 years is not only registered in the sediment layers (first 20 cm, from bottom to top layers), but also reinforced by the Hg accumulation trends found in the annular tree rings (from inner to outer layers). In stations B and C, located away from the most contaminated area, the similarity between sediment and tree trends of Hg accumulation is less noticeable. On the other hand, the majority of the Hg accumulation trends (except tree rings in station C) indicate an increasing level of Hg accumulation in the recent past (10 to 15 years). Even in Pacoti (reference site), where Hg concentration is lower, considering both sediment and tree rings, sediment top layers indicate an increasing Hg accumulation trend in recent times (also pointing to the last 10–15 years). A positive and significant correlation (*p* < 0.05) between Hg trends in sediments and Hg trends in annular tree rings was observed (Figure 4) that corroborates the results registered for Hg in tree ring historical evaluation of Hg patterns of accumulation. This shared pattern of historical Hg accumulation reflects local environmental conditions, linking sediment and tree ring trends of accumulation.

### 3.2. Mercury in Surface Sediments and SPM

Organic matter content in fine sediments (Figure 5a) revealed a statistically significant difference (*p* < 0.05) among sampling sites showing a decreasing gradient towards the sea entrance (from site A to site C, respectively, 25.8 ± 0.6% and 6.8 ± 1.4%) in Ceará River, whereas the reference site Pacoti showed an organic matter content of 10.1 ± 0.9%. In the coarse sediments, the organic matter (Figure 5a) also showed a statistically significant difference among sampling sites (*p* < 0.05), showing very high values (24.0 ± 0.2%) in the upper station of Ceará river (site A) where mangroves are heavily present, whereas lower values were found close to the sea entrance (0.9 ± 0.2% and 2.2 ± 1.2%, respectively, sites B and C) and also in Pacoti (3.1 ± 0.4%) (Figure 5a).

Mercury in fine surface sediments revealed a statistically significant difference (*p* < 0.05) among sampling sites, showing a decreasing gradient towards the sea entrance (from site A to site C, respectively, 62.95 ± 1.56 ng g^−1^ and 12.66 ± 0.41 ng g^−1^) in Ceará River, whereas the reference site Pacoti showed a concentration of 24.15 ± 0.11 ng g^−1^. In the fine sediment fraction Hg concentrations presented statistical differences among sampling sites too (*p* < 0.05), with higher values (62.95 ± 1.56 ng g^−1^) in the upper station of Ceará River (site A) and lower values in the other sampling sites (Figure 5b).

In the SPM, Hg concentrations also presented a statistically significant difference (*p* < 0.05) among sampling sites, showing an increasing gradient in Ceará River, towards the sea entrance (from site A to site C, respectively, 153.1 ± 0.57 ng g^−1^ and 444.8 ± 21.79 ng g^−1^) and a concentration of 660.6 ± 36.37 ng g^−1^ in Pacoti (Figure 5b). The results show that Hg in the surface sediments of Ceará River is especially associated with the fine sediment fraction, mainly due to the increased capacity of small particles to adsorb Hg, given their higher surface area per unit of mass ratio. The analogous Hg concentration found in both sediment fractions in site A may be explained by the high organic matter contents in those fraction particles (Figure 5a). Relating the Hg concentration presented by the fine sediment fraction on Ceará River with the site correspondent SPM, it is possible to see a dissimilar behavior. Whereas Hg concentration on surface sediments decreases from point A to point C (Figure 5b), Hg concentration in SPM increases from point A to point C (Figure 5c).

The geoaccumulation index (Figure 6) indicates that Ceará River presents an increasing upstream contamination gradient, mainly in the sediment top layers, and it also shows an increment in contamination in recent decades in the upstream site A, varying from unpolluted to moderately or even moderately to strongly polluted in the surface sediment layer.

### 3.3. Mercury in Sphoeroides testudineus Tissues

The correlation between the Hg concentration present in the muscle and liver of *S. testudineus* collected in the Pacoti and Ceará rivers and the total length can be seen in Figure 7a,b, respectively. The Hg concentration in muscles obtained in Pacoti River ranged from 32.64 to 97.63 ng g^−1^ and from 37 to 97 ng g^−1^ in those obtained in Ceará River. For liver, the Hg values ranged from 37 to 97 ng g^−1^ in Pacoti River and from 67 to 126 ng g^−1^ in Ceará River.

### 3.4. Biota–Sediment Accumulation Factor Determination

Hg BSAFs found in *S. testudineus* muscles from Pacoti River ranged from 1.35 to 4.04 (Figure 8a) and in Ceará River from 1.43 to 3.70 (Figure 8b). All values indicate that Hg concentrations in fish muscles are higher than concentrations available in surface sediments, indicating that bioaccumulation processes have taken place in both study areas.

Most of the values present in this study have lower [Hg] than that established for sediments and soils, drinking water, biota (EQS), for Water for Protection of Aquatic Life and for human consumption (Table 2), which means that, in terms of risk assessment, the consumption of these species has no negative effects on human health or wildlife. Although, values of Hg concentrations in fish muscles are higher than concentrations available on surface sediments, indicating that bioaccumulation processes have taken place in both study areas.

### 3.5. Conceptual Model of Present Time and Climate Change Scenario

The global results obtained in this study show that, in the northeastern semiarid coastal regions of Brazil, the regional and global drivers, global climate change and some anthropogenic factors, are responsible for a greater mobilization of Hg that allows increasing fluxes and/or bioavailability of this toxic trace element and an acceleration of its biogeochemical transformation. In order to obtain a better visualization and understanding of the Hg dynamic in these aquatic system processes, a conceptual model was developed. This conceptual model represents the attempt to integrate data from these studies and other published data to create a historical summary of the present time and climate change scenario in order to understand processes that control the fate and transport of Hg in this region (Figure 9).

The decreasing continental runoff and increasing estuarine water residence time in this NE Brazilian estuarine system are causing profound changes in Hg fluxes to the ocean and increases in Hg bioavailability. As a consequence, the already meager continental runoff of water has been drastically decreasing and, more recently, a steady decrease in annual rainfall has incremented its effects. In addition, the increase in ocean forcing due to heat accumulation over the continental shelf and further, into the estuary, promotes a greater entry of the saline wedge into the estuarine zone (current), with a consequent retreat (upstream) of the maximum turbidity zone, which strongly affects the hydrodynamics of these aquatic environments. It is a very possible scenario in a mangrove area that would be further invaded by seawater, favoring the incorporation of fine sediments in the seawater column, resulting in a greater mobilization of bioavailable Hg, even due to a greater complexation with chlorides (advancing further upstream), which does allow an acceleration of the biogeochemical transformation of Hg. In these conditions, the threat of heavy metals and non-essential elements to wildlife and humans is exacerbated by their long-term persistence in the environment, which can vary from hundreds to thousands of years. The process of contamination can occur at different levels and due to different modes of exposure (Figure 9).

## 4. Discussion

In the northeastern regions of Brazil, the regional and global drivers responsible for increasing continental runoff and diminishing fluvial discharges to the Western Equatorial South Atlantic Ocean are strengthening due to global warming [56,87,88,89,90]. This region is normally already subject to conditions of semiarid characteristics, but drought events are now intensifying both in terms of their frequency and in the duration of prolonged drought episodes [50,91]. The recent study of Lacerda et al. [56] discusses this topic in detail, the new theory, “Arctic Paradox”, arguing that climate change could be the most important reason for the intensification of this process, increasing the residence time of water in the estuary, and inducing a greater mobilization of bioavailable Hg, which allows an acceleration of the biogeochemical transformation of Hg. In the present work, in order to evaluate the global climate change increase in Hg bioavailability and bioaccumulation in fish and wildlife, chronological Hg contamination trends in the Ceará River estuarine ecosystem, northeastern coast of Brazil, were evaluated, comparing trends in both sediment and tree ring cores. To assess the biological significance of Hg concentrations in sediments to fish and wildlife, muscle and liver from *S. testudineus* were also analyzed. When fish is exposed to Hg contamination, the liver rapidly reflects it, acting in the storage, redistribution and detoxification of this contaminant. The comparison of the top layers of sediment (~20 cm around 15 to 20 years) with the outer layers in the tree cores (planted around 30 years ago) cover a global range of 30 years. Due to its characteristic persistence, Hg is continuously dispersed and the contamination process itself cannot be revealed at the time of its release into the environment [1,2,3,57], which explains the upward-trending, albeit low, concentrations of Hg in sediments and growth rings over the years, visible in the results obtained in Ceará River in this study. The obtained data on historical trends of Hg contamination provide a useful tool for understanding past patterns, evaluating present levels and predicting future evolution and potential intensification of this process due to climate change.

The results regarding Hg in surface sediments and SPM are indicative of the increase in ocean forcing due to heat accumulation over the continental shelf and further, into the estuary, that promotes a greater entry of the saline wedge into the estuarine zone [87]. Similar to what was observed in previous works, these processes strongly affect the hydrodynamics of these aquatic environments [87,88,89,90], inducing maximum turbidity zone retreat and favoring the incorporation of fine sediments into the seawater column [88,92]. Climate change could be the most important reason for the intensification of these processes and is a very possible scenario in a mangrove area such as the Ceará River estuary, that would be further invaded by seawater, resulting in a greater mobilization of bioavailable Hg, even due to a greater complexation with chlorides (advancing further upstream), which does allow an acceleration of the biogeochemical transformation of Hg [56]. The higher Hg concentration on SPM than the correspondent surface sediment suggests that Hg-contaminated suspended particles may be entering the Ceará River estuary from the coast area of the Metropolitan Region of Fortaleza (MRF) where, according to the research of Marins et al. [93], there is a sewer emissary of urban and domestic waste. Those contaminated particles may enter the river estuary suspended by the saltwater, settling as they encounter freshwaters, accumulating especially in sampling site A. The results indicate that a similar phenomenon could be happening in the less impacted Pacoti estuary. The Ceará River serves as a receptor of the domestic sewage and local discharge of the effluents industries of Fortaleza, and also from the industries of Maracanaú through Maranguapinho River, which flows into it [75,94]. One explanation for the rise in Hg concentration could be the placement of the outfall of Fortaleza (ESF) since 1978, because besides the high dilution capacity of contaminants to the ocean, the river showed contaminated areas around it, dispersing in the east–west [94]. The Hg levels found in the fine sediments fraction of the coast of Fortaleza Metropolitan Region ranged from 0.72 to 17.54 ng g^−1^, with a maximum concentration at the point corresponding to output of the emissary [94]. At this point, the Hg content was slightly higher than the geochemical background for the east coast of Brazil, which is 15 ng g^−1^ for the fine sediments fraction. Additionally, in the port area (5.50 ng g^−1^) and the mouth of the Ceará River (7.13 and 3.23 ng g^−1^), the concentrations are high, denoting the influence of the river on emissions of Hg [94]. Values reported to the metropolitan region of Fortaleza set background values of the estuary of Ceará of 1 to 10 ng g^−1^ [93]. These previous studies indicate the strong influence of population and industrial development, with greater emphasis on the outfall, but also giving importance to the Ceará River, which receives all effluent, and also the Port of Mucuripe. The obtained results, varying from unpolluted outer zone of the estuary to moderately polluted inner zone, are similar to other previous works in the region. Despite the increase in Hg in recent years, current Hg concentrations in Ceará River estuary are relatively low when compared with sites from other parts of the world. For example, Rio Grande do Sul, Brazil [95]; Rio de Janeiro, Brazil [96]; Asturias, Spain [97]; Estarreja Channel, Aveiro, Portugal [36]; Krka River estuary, Croatia [98]; Conwy estuary, Wales [99]; Marano and Grado Lagoon, Italy [100]; Thames Estuary, London, UK [101]; and Aveiro Lagoon, Aveiro, Portugal [102]. The obtained results, varying from unpolluted outer zone of the estuary to moderately polluted inner zone, are similar to other previous works in the region [51,93,94,103]. However, surface sediments from Beira-Mar Beach, located near the sewer emissary of urban and domestic waste on the coast area of the Metropolitan Region of Fortaleza, presented more than double the Hg concentration found on the estuary, indicating that the emissary can be the contamination source [56,76]. The results of the geoaccumulation index may indicate that these processes are inducing a greater mobilization of bioavailable Hg, which could allow an acceleration of the biogeochemical transformation of Hg [56]. Mercury patterns reflect local environmental conditions of this estuarine area, that beyond the anthropogenic pressures, are also suffering the influence of global drivers associated with the increasing continental runoff and diminishing fluvial discharges to the Western Equatorial South Atlantic Ocean and further invasion by seawater, which could favor the incorporation of fine sediments into the seawater column [88,92].

The assessment of the biological significance of Hg concentrations in sediments to fish and wildlife showed that fish muscle appears to act mostly as a reservoir, accumulating the Hg in order to protect other organs [104]. As liver is the most common tissue for monitoring environmental contaminant exposure in fish [105,106], researchers have attempted to establish toxicity thresholds for contaminants based on a consideration of the lowest liver concentrations that are associated with significant toxic effects in individual animals. The threshold is 25–30 μg g^−1^ ww Hg in liver, above which animals are likely to experience MeHg intoxication and death [35]. It is expected that most of the Hg in the dosed animals’ tissues was present as MeHg. Unfortunately, without additional information on the proportions of MeHg and inorganic Hg in the liver, a toxicity threshold expressed solely on a total Hg basis is insufficient for making confident toxicological assessments. The Hg concentration values in *S. testudineus* from both study areas are not restrictive to human consumption, being below the legislated European limit for Hg in foodstuffs. Additionally, most of the values present in this study have lower [Hg] than that established for sediments and soils, drinking water, biota (EQS), for Water for Protection of Aquatic Life and for human consumption, which means that, in terms of risk assessment, the consumption of these species has no negative effects on human health or wildlife. Although, values of Hg concentrations in fish muscles are higher than concentrations available in surface sediments, indicating that bioaccumulation processes have taken place in both study areas. The results from *S. testudineus* muscles analysis suggest a significant and linear increase in Hg burden with increasing fish length, indicating that the specimens accumulate Hg as they grow. The concentrations found in the *S. testudineus* livers corroborate this hypothesis, since a higher concentration of Hg in liver is a primary indication of fish exposure to the contaminant [107,108]. The range of toxic effects in fish at environmentally relevant levels of MeHg exposure indicates that changes in biochemical processes, damage to cells and tissues and reduced reproduction in fish begin to occur at concentrations of about 0.5–1.0 μg Hg g^−1^ ww in axial muscle (>90% of Hg in muscle is MeHg) [109]. The results of fish toxicology studies, estimated with a mathematical model, show the lowest observable adverse effects level (LOAEL) of about 0.3 μg Hg g^−1^ ww in the whole body of fish, or about 0.5 μg Hg g^−1^ ww in axial muscle [109]. Although, other climate-related factors would likely indirectly affect contaminant bioaccumulation in this resident fish. Climate change encompasses multiple changes in physical factors (wind flows, atmospheric circulation and precipitation) that will likely affect Hg deposition and bioavailability, and biological factors (species biogeographic patterns, composition) that affect food web structure and in turn bioaccumulation [110]. In the context of the Ceará River estuary, changes such as those in nutrient loading in estuaries due to changes in precipitation could also alter estuarine trophic status, thereby changing MeHg production and bioaccumulation [2]. Considering that when exposed to a steady concentration of Hg in the environment fish tend to present a ratio [Hg] in liver/[Hg] in muscles = 1, indicating a state of quasi equilibrium with the environment, a [Hg] in liver/[Hg] in muscles ratio >1 indicates that the *S. testudineus* from both study areas are experiencing an increase in Hg bioavailability [56]. *S. testudineus* being a species of benthic habits, oriented toward the surface sediments, the accumulation factor between the specimens’ organs and the associated sediments enables us to determine their ability to accumulate the Hg presently available in the study areas [84]. The results from both rivers show an increase in BSAF with fish growth; this may be explained by the ability of fish to forage on bigger prey, with more time of exposure to the contaminant.

Mercury exposure is a global health threat to ecosystems and wildlife that can affect multiple organ systems, and manifests as diverse adverse health outcomes in wildlife and humans, particularly within the context of rapid environmental global changes [1,2,6,7,8]. In this context, managing aquatic systems is becoming increasingly complex due to human impacts, multiple and competing water needs and climate variability [4,45,47,111]. Possible climate-induced shifts in aquatic systems highlight the need for accurate and regionally specific metrics of change in the past in response to climate, and for improved understanding of responses to climate factors [41,47,48,111]. The results of this work reinforce the indicators previously described in the semiarid NE region of Brazil [50,56,91], which showed that global climate change and some anthropogenic factors are key drivers of Hg exposure and biomagnification for wildlife and humans. The uptake, metabolism and effects of Hg in fish and wildlife have been studied for more than 50 years, and much has been learned regarding the dynamics of Hg in the environment, its food chain transfer and its toxic effects [2,3,38,41]. Recent climate change scenarios for aquatic resources highlight uncertainties about the timing and magnitude of river flows that could lead to escalating ecosystem stress [112]. These widespread environmental and health questions are essential to ecosystem, wildlife and human health, and they deserve international attention given the relevance of understanding Hg remobilization processes to social and economic development and their response to global environmental changes. There is a critical need to understand further the interactions of negative anomalies in annual rainfall and continental runoff reduction to better predict changes in exposure trajectories and realize adverse, potentially important health effects of Hg on behavior, neurochemistry and endocrine function in fish and wildlife at currently realistic levels of environmental exposure. More information on the impacts of effluents quality on the chemical composition of tree rings could be a useful monitoring tool to evaluate the spatial and temporal patterns of effluents use and would further contribute to understanding and identifying high-frequency changes in surface water quality patterns. Future research will be important to monitor and regulate Hg levels in aquatic systems to prevent Hg reaching human populations through seafood consumption and reduce Hg emission to the environment, which can make a significant difference in recovering the world’s ecosystems from Hg pollution, leaving a minimum legacy of Hg pollution in the ocean and aquatic systems for our future generations.

## 5. Conclusions

Managing aquatic systems is becoming increasingly complex due to human impacts, multiple and competing water needs and climate variability. The results of this work reinforce the indicators previously described in the semiarid NE region of Brazil, which showed that global climate change and some anthropogenic factors are key drivers of Hg exposure and biomagnification for wildlife and humans. Considering the Hg concentration present in the top layers of sediment (~20 cm around 30 to 40 years) with the outer layers in the tree ring cores and in the sediment cores from Pacoti estuary and the Ceará estuary, overall the data indicate an increase in Hg in recent years. A positive and significant correlation (*p* < 0.05) was revealed between Hg trends in sediments and Hg trends in annular tree rings. This shared Hg pattern reflects local environmental conditions. The results show that Hg in the surface sediments is especially associated with the fine sediment fraction, mainly due to the increased capacity of small particles to adsorb Hg, given their higher surface area per unit of mass ratio.

The Biota–Sediment Accumulation Factor indicates that Hg concentrations in fish muscles are higher than concentrations available on surface sediments, leading to the conclusion that bioaccumulation processes have taken place in both study areas. The muscle appears to act mostly as a reservoir, accumulating the Hg. Although, as liver is the most common tissue for monitoring environmental contaminant exposure in wildlife, it is expected that most of the Hg in the dosed animals’ tissues was present as MeHg. Hg in the liver, a toxicity threshold expressed solely on a total Hg basis, is insufficient for making confident toxicological assessments. The Hg concentration values in *S. testudineus* from both study areas are not restrictive to human consumption, being below the legislated European limit for Hg in foodstuffs. Although, results from *S. testudineus* muscles analysis suggest a significant and linear increase in Hg burden with increasing fish length, indicating that the specimens accumulate Hg as they grow. The results from both rivers show an increase in BSAF with fish growth; this may be explained by the ability of fish to forage on bigger prey, with more time of exposure to the contaminant. Considering that when exposed to a steady concentration of Hg in the environment, fish tend to present a ratio [Hg] liver/[Hg] muscles = 1, indicating a state of quasi equilibrium with the environment, the [Hg] liver/[Hg] muscles ratio >1 indicates that the *S. testudineus* from both study areas are experiencing an increase in Hg bioavailability. Possible climate-induced shifts in these aquatic systems highlight the need for accurate and regionally specific metrics of change in the past in response to climate and for improved understanding of response to climate factors. These processes are inducing a greater mobilization of bioavailable Hg, which could allow an acceleration of the biogeochemical transformation of Hg.

## Figures and Tables

**Figure 1 animals-11-02402-f001:**
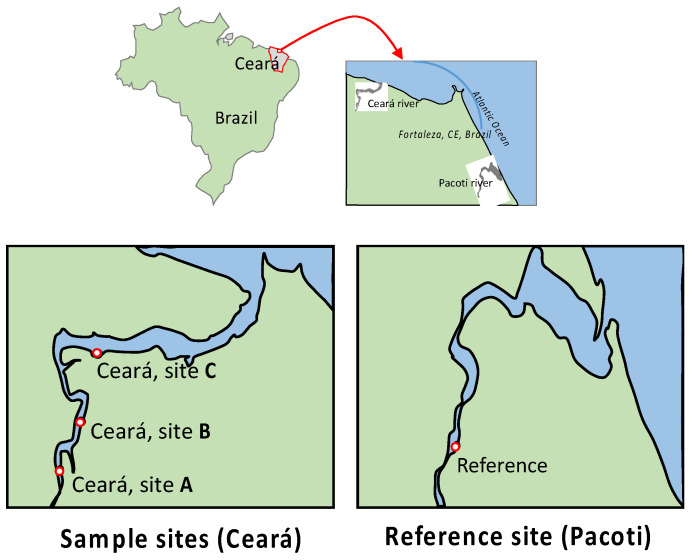
Location of sampling sites along Ceará estuary. Ceará A (3°43′11,06″ S 38°37′22,4″ W, near the spring of Ceará River); Ceará B (3°42′59,1″ S 38°37′16,75″ W, local reception of Maranguapinho River); Ceará C (3°42′73″ S 38°36′52,73″ W, near the mouth of the Ceará River); Pacoti (3°49′52,2″ S 38°25′11,78″ W, local reference).

**Figure 2 animals-11-02402-f002:**
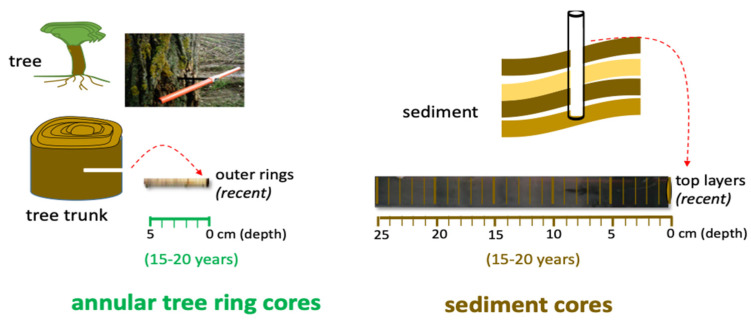
Study sampling design that includes an application of dendrochemistry analysis (annular tree rings analysis), through registration of Hg concentration on growth rings in specimens of *Rhizophora mangle* L. and using the assessment in sediments as a support for comparison of profiles of contamination.

**Figure 3 animals-11-02402-f003:**
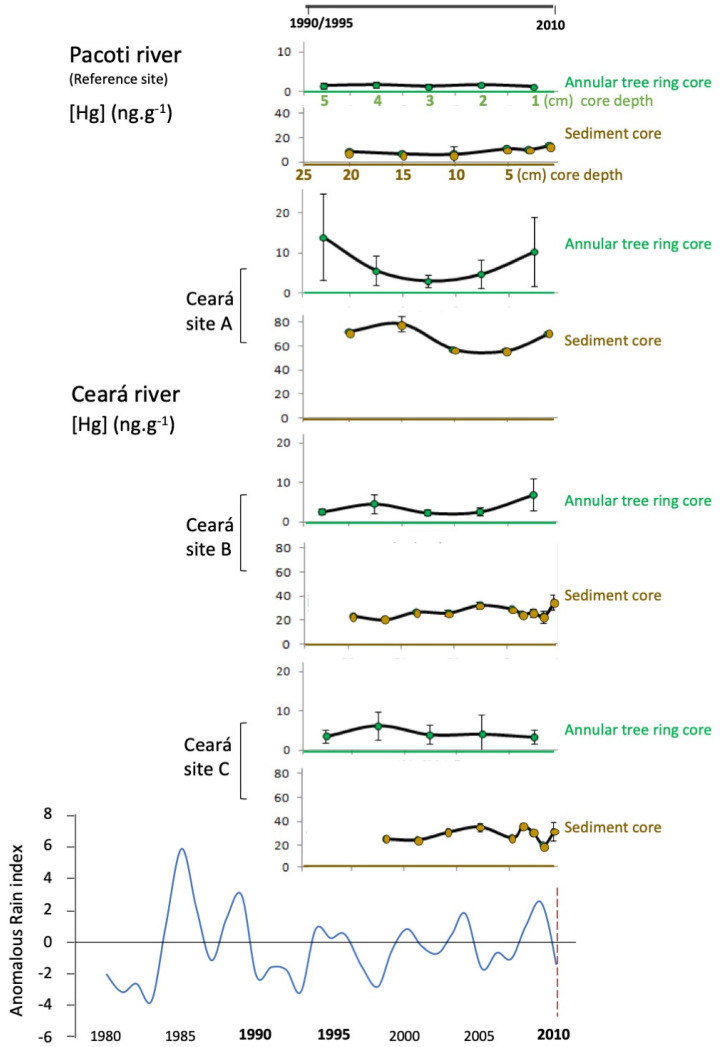
Sediment cores (surface sediments, meaning 0–2 cm removable bed particles) and annular tree rings surveyed in Pacoti (the reference site) and in Ceará estuary (three sites, A, B and C, downstream of the estuary), portraying the chronological pattern of Hg accumulation, during the last 15–20 years. Historical pluviometry adapted from FUNCEME (1990–2014).

**Figure 4 animals-11-02402-f004:**
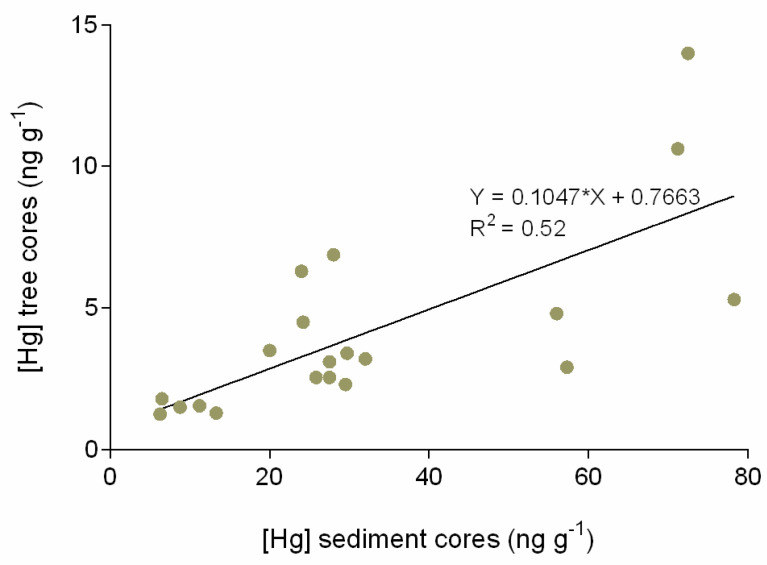
Dispersion graphic of Hg comparing Hg concentration in sediment cores versus Hg concentration in tree ring cores.

**Figure 5 animals-11-02402-f005:**
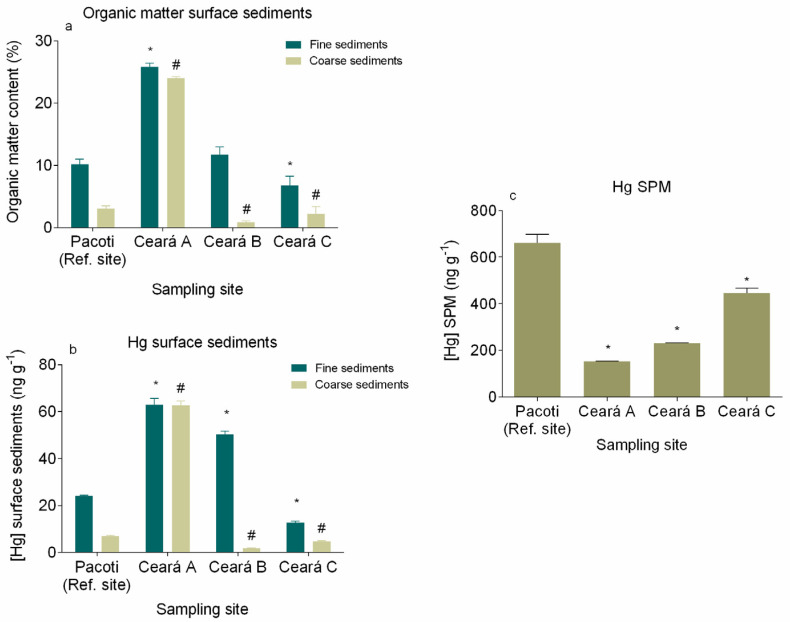
Organic matter content (%) (**a**), Hg concentration in surface sediments (fine and coarser sediments) (**b**) and Hg concentration from the different sampling sites (**c**). * denotes a significant difference (*p* < 0.05) in fine sediments and SPM when compared with the reference site (Pacoti) and # denotes a significant difference (*p* < 0.05) in coarser sediments when compared with the reference site (Pacoti).

**Figure 6 animals-11-02402-f006:**
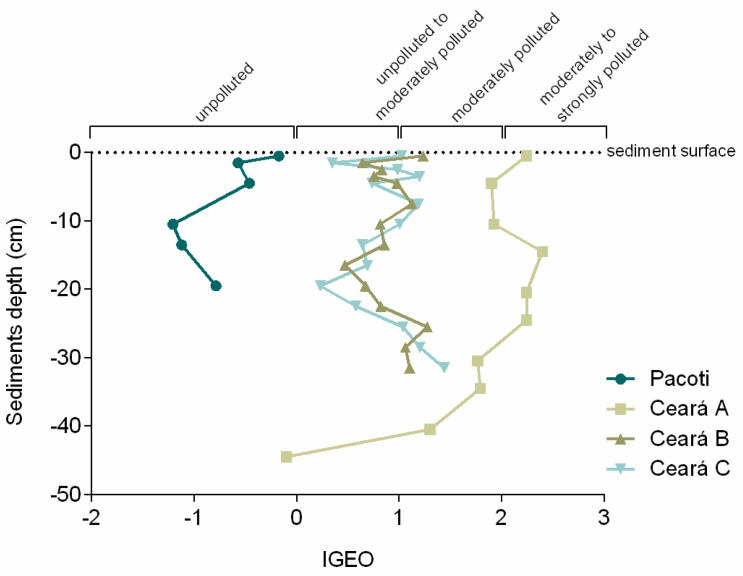
Igeo (value and class) for each sampling site.

**Figure 7 animals-11-02402-f007:**
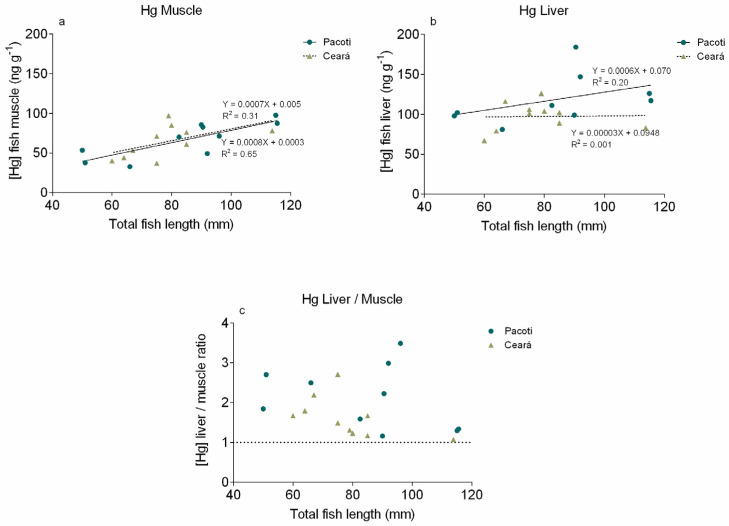
Specimen length vs. Hg concentration in *S. testudineus* muscle (**a**) and liver (**b**) from Pacoti and Ceará and the ratio between Hg concentration present in liver and muscle (**c**) in both study areas (Pacoti and Ceará).

**Figure 8 animals-11-02402-f008:**
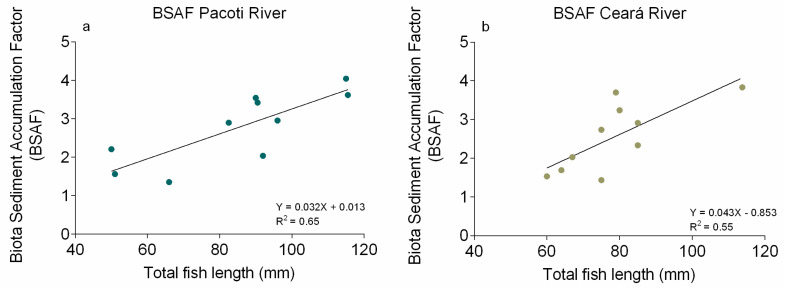
Biota Sediment Accumulation Factor (BSAF) between [Hg] in *S. testudineus* specimens’ muscle and in the associated sediments from Pacoti river (**a**) and Ceará river (**b**).

**Figure 9 animals-11-02402-f009:**
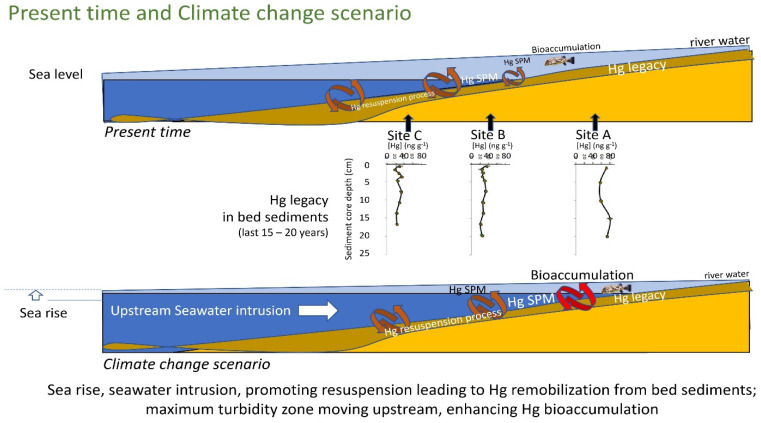
Conceptual model integrating data from these studies and other published data to create a historical summary of the present time and climate change scenario of Hg mobilization fluxes and/or bioavailability and its biogeochemical transformation.

**Table 1 animals-11-02402-t001:** Geoaccumulation index (Igeo) in relation to pollution intensity.

Igeo	Class	Pollution Intensity
<0	1	unpolluted
0–1	2	unpolluted to moderately polluted
1–2	3	moderately polluted
2–3	4	moderately to strongly polluted
3–4	5	strongly polluted
4–5	6	strongly to very strongly polluted
>5	7	very strongly polluted

**Table 2 animals-11-02402-t002:** Average Hg concentration (µg g^−1^) obtained in this study (sediments, fish tissues and Biota–Sediment Accumulation Factor determination (BASF)) and the maximum Hg concentrations established by international environmental guidelines standards to protect wildlife (EQS) and humans from health problems (numeric criteria for maximum acceptable concentration of Hg Baseline Values Standard for Soil and Sediments, marine water, Water for Protection of Aquatic Life, fish tissue, drinking water and for human consumption.

International Hg Environmental Guidelines	Hg Source	[Hg] (µg g^−1^)	Guidelines Standards
EQS_Biota_ ^(1)^	-	0.02	
Limit for human consumption ^(2)^	-	0.5	EPA ^(3)^; FDA ^(4)^
Limit of Mercury for Water for Protection of Aquatic Life	Inorganic Hg Methyl mercury	0.026 0.004	EPA ^(3)^
Limit of Mercury for Fish ^(4)^		1	FDA action level for methyl mercury
Drinking water	-	1 mg/L (total Hg) 2 mg/L (Inorganic Hg)	WHO ^(9)^; EPA
Baseline Values Standard for Soil and Sediments	Marine water	0.13	ISQG ^(7)^
Baseline Values Standard for Soil and Sediments	Baseline	0.4	World Shale Value (µg/g) ^(6)^
Baseline Values Standard for Soil and Sediments	Marine water	0.70	PEL ^(8)^
Baseline Values Standard for Soil and Sediments (Soils)	Residential	6.6	Guideline value EPA, WHO ^(5)^
Baseline Values Standard for Soil and Sediments (Soils)	Industrial	50	Guideline value EPA, WHO
Baseline Values Standard for Soil and Sediments (Soils)	Agricultural	6.6	Guideline value EPA, WHO
Pacoti River (Ceará, Brazil)	Fine sediments (<63 µm)	0.02–0.02	This study
Ceará River (Ceará, Brazil)	Fine sediments (<63 µm)	0.01–0.07	This study
Pacoti River (Ceará, Brazil)	Hg in muscles (*S. testudineus*)	0.03–0.10	This study
Ceará River (Ceará, Brazil)	Hg in muscles (*S. testudineus*)	0.03–0.10	This study
Pacoti River (Ceará, Brazil)	Hg in liver *S. testudineus*)	0.08–0.18	This study
Ceará River (Ceará, Brazil)	Hg in liver (*S. testudineus*)	0.07–0.13	This study
BASF—Pacoti River (Ceará, Brazil)		1.35–4.04	This study
BASF—Ceará River (Ceará, Brazil)		1.43–3.70	This study

^(1)^ Directive 2013/39/EU; ^(2)^ Commission Regulation (EC) No 1881/2006; ^(3)^ The United States Environmental Protection Agency (USEPA) regulates mercury in pesticides, and mercury release into the environment through air, water and land disposal limits. ^(4)^ The US Food and Drug Administration (FDA) regulates mercury in cosmetics, food and dental products. ^(5)^ Guideline Value EPA, WHO; ^(6)^ World Shale Value EPA, WHO; ^(7)^ ISQG—Interim Sediment Quality Guideline; ^(8)^ PEL—Probable Effect Level; ^(9)^ WHO—World Health Organization.

## Data Availability

The data presented in this study are available in the current manuscript; raw data are available on request from the corresponding author.

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
