# Peer review of "Chronological Trends and Mercury Bioaccumulation in an Aquatic Semiarid Ecosystem under a Global Climate Change Scenario in the Northeastern Coast of Brazil"

_animals, 2021, doi:10.3390/ani11082402_

Round 1

Reviewer 1 Report

Chronological Trends and Bioaccumulation of Mercury in a Global Climate Change Scenario (Aquatic Semiarid Ecosystem, North-Eastern Coast of Brazil)

This paper presents valuable data. However, there are several details that the authors must consider and improve their manuscript. The major problem is that the Introduction is too long and lacks elucidation of important aspects (point 4 below). Another problem is that the Results and Discussion are not clearly marked. The results section includes references which should not be the case. The writing is verbose and can be condensed significantly (reduced by 30%) which will improve our understanding of the text.  There are too many grammatical errors. In just the abstract I found more than 10 errors. Please have the entire text revised by a native English speaker.

Specific comments

  1. I do not find the simple summary in the document.
  2. Are we at zero Hg discharge? What about Hg from the burning of fossil fuels? Please clarify and correct.
  3. Lines 133-137 are not clear and can be deleted.
  4. The introduction does not present clearly as to why this study included trees on one hand and fish on the other. Please explain.
  5. Why do we find references in the results section? Are the authors presenting results and discussion together? Please delete the references in the results.
  6. Lines 308-333 read less like results and more as discussion or introduction. Please modify.
  7. 2 not clear since it refers to depth and chronology-please modify the graph.
  8. Please add more information and comparisons on/with other species of fish.
  9. Line 404 onwards reads like the Discussion. Please highlight with appropriate heading to the sub-section.

Author Response

Fernando Morgado

CESAM, Department of Biology

University of Aveiro,

3810-193 Aveiro,

Portugal

30th July 2021

Dear Editor,

The authors would like to thank your considerations on the manuscript entitled “Chronological trends and mercury bioaccumulation of mercury in an aquatic semiarid ecosystem under a global climate change scenario in the North-eastern coast of Brazil”, sent for publication to Animals.

We have taken in consideration all the comments and suggestions from the reviewers and revised the manuscript accordingly.The article has been substantially improved. Following the reviewers' suggestions for corrections, an extensive review of the article was carried out. In addition to a spelling and grammar review, all the recommended structural changes were carried out: inclusion of a simple summary, abstract rewritten and enhanced with the inclusion of the suggested information, substantial reduction of the introduction (more or less 30%), separation of the results and discussion and the inclusion of a conclusion section. One new figure was also included in the materials and methods section in order to a better understanding of the study sampling design, and to increase the understanding of the texts. One new table was also included in the results containing the average Hg concentration (µg g-1) obtained in this study (sediments, fish tissues and Biota-Sediment Accumulation Factor determination (BASF)) and the maximum Hg concentrations established by international environmental guidelines standards to protect wildlife (EQS) and humans from health problems (numeric criteria for maximum acceptable concentration for Hg Baseline Values Standard for Soil and Sediments, marine water, Water for Protection of Aquatic Life, fish tissue, drinking water, and for human consumption. This table increases the ability to interpret the results in terms of risk assessment and negative effects on wildlife or human health. One new figure was also included in the results section, a conceptual model, in order to take a better visualization and understanding of the Hg dynamic on these aquatic system processes..This conceptual model represents the attempt to integrate data from these study and other published data to create an historical summary of present time and Climate Change scenario in order to understand processes that control the fate and transport of Hg in this region.

Thank you for the opportunity to improve the paper.

We hope that you will find the revised manuscript suitable for publication.

Below is our response to reviewers’ comments.

With my best regards,

Fernando Morgado (on behalf of all authors)

Reviewer #1

This paper presents valuable data. However, there are several details that the authors must consider and improve their manuscript. The major problem is that the Introduction is too long and lacks elucidation of important aspects (point 4 below). Another problem is that the Results and Discussion are not clearly marked. The results section includes references which should not be the case. The writing is verbose and can be condensed significantly (reduced by 30%) which will improve our understanding of the text.  There are too many grammatical errors. In just the abstract I found more than 10 errors. Please have the entire text revised by a native English speaker.

Authors: Following the reviewers' suggestions for corrections, an extensive review of the article was carried out. In addition to a spelling and grammar review, all the recommended structural changes were carried out: inclusion of a simple summary, abstract rewritten and enhanced with the inclusion of the suggested information, substantial reduction of the introduction (more or less 30%), separation of the results and discussion and the inclusion of a conclusion section. One new figure was also included in the materials and methods section in order to a better understanding of the study sampling design, and to increase the understanding of the texts. One new table was also included in the results containing the average Hg concentration (µg g-1) obtained in this study (sediments, fish tissues and Biota-Sediment Accumulation Factor determination (BASF)) and the maximum Hg concentrations established by international environmental guidelines standards to protect wildlife (EQS) and humans from health problems (numeric criteria for maximum acceptable concentration for Hg Baseline Values Standard for Soil and Sediments, marine water, Water for Protection of Aquatic Life, fish tissue, drinking water, and for human consumption. This table increases the ability to interpret the results in terms of risk assessment and negative effects on wildlife or human health.  One new figure was also included in the results section, a conceptual model, in order to take a better visualization and understanding of the Hg dynamic on these aquatic system processes.. This conceptual model represents the attempt to integrate data from these study and other published data to create an historical summary of present time and Climate Change scenario in order to understand processes that control the fate and transport of Hg in this region

  1. I do not find the simple summary in the document.

Authors: A simple summary was included

       2.Are we at zero Hg discharge? What about Hg from the burning of fossil fuels? Please clarify and correct.

Authors: The sentence was deleted

  1. Lines 133-137 are not clear and can be deleted.

Authors:The sentence was deleted

  1. The introduction does not present clearly as to why this study included trees on one hand and fish on the other. Please explain.

Authors: Some texts were introduced in order to better detail the methodological narrative (sediment colors, tree rings colors fish tissues)

  1. Why do we find references in the results section? Are the authors presenting results and discussion together? Please delete the references in the results.

Authors: The separation of the results and discussion sections and the inclusion of a conclusions section were carried out.

  1. Lines 308-333 read less like results and more as discussion or introduction. Please modify.

Authors: The separation of the results and discussion sections and the inclusion of a conclusions section were carried out. A separation of the results and discussion sections was carried out. One new figure was also included in the results section, a conceptual model, in order to take a better visualization and understanding of the Hg dynamic on these aquatic system processes.. This conceptual model represents the attempt to integrate data from these study and other published data to create an historical summary of present time and Climate Change scenario in order to understand processes that control the fate and transport of Hg in this region. Also the inclusion of a much more oriented discussion section was carried out,

  1. Figure 2 not clear since it refers to depth and chronology-please modify the graph.

Authors: Figure 2 has been modified in order to provide more detailed information and easier interpretation. In addition, and in order to complement the information included in figure 2 and the results obtained within the narrative inherent to the objectives of the article, a new figure was included in the article.

  1. Please add more information and comparisons on/with other species of fish.

Authors: One new table was also included in the results containing the average Hg concentration (µg g-1) obtained in this study (sediments, fish tissues and Biota-Sediment Accumulation Factor determination (BASF)) and the maximum Hg concentrations established by international environmental guidelines standards to protect wildlife (EQS) and humans from health problems (numeric criteria for maximum acceptable concentration for Hg Baseline Values Standard for Soil and Sediments, marine water, Water for Protection of Aquatic Life, fish tissue, drinking water, and for human consumption. This table increases the ability to interpret the results in terms of risk assessment and negative effects on wildlife or human health.

  1. Line 404 onwards reads like the Discussion. Please highlight with appropriate heading to the sub-section.Interesting and relevant research, but I have some notes to presentation and material.

Authors: The separation of the results and discussion sections and the inclusion of a conclusions section were carried out. A separation of the results and discussion sections was carried out. One new figure was also included in the results section, a conceptual model, in order to take a better visualization and understanding of the Hg dynamic on these aquatic system processes.. This conceptual model represents the attempt to integrate data from these study and other published data to create an historical summary of present time and Climate Change scenario in order to understand processes that control the fate and transport of Hg in this region. Also the inclusion of a much more oriented discussion section was carried out.

Reviewer 2 Report

Comments and suggestions

In addition to the corrections suggested in-text, the following additional comments have been made.

Title:

Too broad and does not depict the scope of the study.

Suggestion: Chronological Trends and Bioaccumulation of Mercury in a Global Climate Change Scenario of the North-Eastern Coast of Brazil

Abstract:

Poorly written. The abstract made attempt at presenting the objective of the study and a brief methodology. The design of the study was not however, mentioned. The results obtained from the study were not presented and a conclusion was also lacking in the abstract.

Introduction:

  • The introduction is too long. Consider reducing the length.
  • The introduction was poorly written and will need a major English language editing.
  • The authors’ style of writing includes very long sentences that make it difficult for the reader to follow through. Punctuation marks were not properly used, thereby making comprehension less easy.

Methodology:

Although the authors made a good attempt at detailing the materials and methodologies used, a detailed description of the study design and how it intends to answer the questions raised by the objectives have not been provided. Additionally, the use of long sentences with unclear meanings have undermined the readers understanding and appreciation of the concept of the study.

Results and Discussion

The authors should indicate if this section is to be considered to have included discussions as it is lacking in the title. The authors made a good attempt to present the results of the studies while speaking to it. However, the objective of the study which entails “evaluating the chronological patterns of Hg contamination as useful tool for evaluating past patterns, understanding present levels and predicting future evolution trend in a Global Climate Change scenario” has not been tackled properly. While the authors presented figures showing a trend in the various Hg indicators over some time, the temporal specificities were not duly presented. Also, there were no serious attempts by the authors to give details of this chronological changes as indicated by the dendrochemical studies. In other words, the expectations of seeing how the dendrochemical studies could evaluate the past, show the present, and project the future contamination status of Hg, and its implication for defining the climate changes impacts were cut short. Further, it was evident that the authors are not fluent in English writing, a major factor that contributed to the poor understanding and appreciation of the text.

In general, the study is good one, and has a lot of prospects particularly in the face of the continued climate change footprint on the environmental, animal and human health. It promises great readership to a varied class of readers. However, it is poorly written, and indicates the need for the authors to put in great deal of works if the manuscript could be considered favorably.

Author Response

Fernando Morgado

CESAM, Department of Biology

University of Aveiro,

3810-193 Aveiro,

Portugal

30th July 2021

Dear Editor,

The authors would like to thank your considerations on the manuscript entitled “Chronological trends and mercury bioaccumulation of mercury in an aquatic semiarid ecosystem under a global climate change scenario in the North-eastern coast of Brazil”, sent for publication to Animals.

We have taken in consideration all the comments and suggestions from the reviewers and revised the manuscript accordingly.The article has been substantially improved. Following the reviewers' suggestions for corrections, an extensive review of the article was carried out. In addition to a spelling and grammar review, all the recommended structural changes were carried out: inclusion of a simple summary, abstract rewritten and enhanced with the inclusion of the suggested information, substantial reduction of the introduction (more or less 30%), separation of the results and discussion and the inclusion of a conclusion section. One new figure was also included in the materials and methods section in order to a better understanding of the study sampling design, and to increase the understanding of the texts. One new table was also included in the results containing the average Hg concentration (µg g-1) obtained in this study (sediments, fish tissues and Biota-Sediment Accumulation Factor determination (BASF)) and the maximum Hg concentrations established by international environmental guidelines standards to protect wildlife (EQS) and humans from health problems (numeric criteria for maximum acceptable concentration for Hg Baseline Values Standard for Soil and Sediments, marine water, Water for Protection of Aquatic Life, fish tissue, drinking water, and for human consumption. This table increases the ability to interpret the results in terms of risk assessment and negative effects on wildlife or human health. One new figure was also included in the results section, a conceptual model, in order to take a better visualization and understanding of the Hg dynamic on these aquatic system processes..This conceptual model represents the attempt to integrate data from these study and other published data to create an historical summary of present time and Climate Change scenario in order to understand processes that control the fate and transport of Hg in this region.

Thank you for the opportunity to improve the paper.

We hope that you will find the revised manuscript suitable for publication.

Below is our response to reviewers’ comments.

With my best regards,

Fernando Morgado (on behalf of all authors)

Reviewer #2

In addition to the corrections suggested in-text, the following additional comments have been made.

Authors: Following the reviewers' suggestions for corrections, an extensive review of the article was carried out. In addition to a spelling and grammar review, all the recommended structural changes were carried out: inclusion of a simple summary, abstract rewritten and enhanced with the inclusion of the suggested information, substantial reduction of the introduction (more or less 30%), separation of the results and discussion and the inclusion of a conclusion section. One new figure was also included in the materials and methods section in order to a better understanding of the study sampling design, and to increase the understanding of the texts. One new table was also included in the results containing the average Hg concentration (µg g-1) obtained in this study (sediments, fish tissues and Biota-Sediment Accumulation Factor determination (BASF)) and the maximum Hg concentrations established by international environmental guidelines standards to protect wildlife (EQS) and humans from health problems (numeric criteria for maximum acceptable concentration for Hg Baseline Values Standard for Soil and Sediments, marine water, Water for Protection of Aquatic Life, fish tissue, drinking water, and for human consumption. This table increases the ability to interpret the results in terms of risk assessment and negative effects on wildlife or human health. One new figure was also included in the results section, a conceptual model, in order to take a better visualization and understanding of the Hg dynamic on these aquatic system processes.. This conceptual model represents the attempt to integrate data from these study and other published data to create an historical summary of present time and Climate Change scenario in order to understand processes that control the fate and transport of Hg in this region

  1. Title: Too broad and does not depict the scope of the study. Suggestion: Chronological Trends and Bioaccumulation of Mercury in a Global Climate Change Scenario of the North-Eastern Coast of BrazilI

Authors: The title has been changed according to the reviewers in order to be more within the scope of the study

  1. do not find the simple summary in the document.

Authors: A simple summary was included

  1.  Abstract: Poorly written. The abstract made attempt at presenting the objective of the study and a brief methodology. The design of the study was not however, mentioned. The results obtained from the study were not presented and a conclusion was also lacking in the abstract.

      Authors: The abstract has been rewritten and enhanced with the inclusion of suggested information

  1. Introduction: i) The introduction is too long. Consider reducing the length. ii) The introduction was poorly written and will need a major English language editing. iii) The authors’ style of writing includes very long sentences that make it difficult for the reader to follow through. Punctuation marks were not properly used, thereby making comprehension less easy.

Authors: An extensive review of the introduction was carried out which included a substantial reduction of the the length (more or less 30%) and a spelling and grammatical revision

Methodology: Although the authors made a good attempt at detailing the materials and methodologies used, a detailed description of the study design and how it intends to answer the questions raised by the objectives have not been provided. Additionally, the use of long sentences with unclear meanings have undermined the readers understanding and appreciation of the concept of the study.

Authors: A spelling and grammatical revision was carried out. Also a more detailed description of the study design and how it intends to answer the questions raised by the objectives was carried out with the inclusion of more detailed texts and also a new figure in order to clarify the study design and how it intends to answer the questions raised by the objectives.

Results and Discussion: The authors should indicate if this section is to be considered to have included discussions as it is lacking in the title. The authors made a good attempt to present the results of the studies while speaking to it. However, the objective of the study which entails “evaluating the chronological patterns of Hg contamination as useful tool for evaluating past patterns, understanding present levels and predicting future evolution trend in a Global Climate Change scenario” has not been tackled properly. While the authors presented figures showing a trend in the various Hg indicators over some time, the temporal specificities were not duly presented. Also, there were no serious attempts by the authors to give details of this chronological changes as indicated by the dendrochemical studies. In other words, the expectations of seeing how the dendrochemical studies could evaluate the past, show the present, and project the future contamination status of Hg, and its implication for defining the climate changes impacts were cut short. Further, it was evident that the authors are not fluent in English writing, a major factor that contributed to the poor understanding and appreciation of the text.

Authors: The separation of the results and discussion sections, the inclusion of a conclusions section and a spelling and grammatical revision were carried out. One new figure was also included in the results section, a conceptual model, in order to take a better visualization and understanding of the Hg dynamic on these aquatic system processes.. This conceptual model represents the attempt to integrate data from these study and other published data to create an historical summary of present time and Climate Change scenario in order to understand processes that control the fate and transport of Hg in this region. Also the inclusion of a much more oriented discussion section was carried out.

Reviewer 3 Report

The topic is really interesting, however the manuscript needs many corrections and improvements before getting published. Some data (Figures) are missing, so it is impossible to check if the description is correct. You should carefully check all figures and adjust the text accordingly. Conclusions which answer the question in hypothesis must be provided.

Detailed suggestion:

Line 218 – different way of referencing to the literature than in the rest of the text – it should be as number not as the name of author.

Line 222 – does it mean that there were three trees in each site?

Line 226- 229 The sentence “Trees were planted in the 1980 decade, meaning that the older trees have about 30 years and showing about 10cm radial growing from the tree center, thus each centimeter radial segment corresponds to approximately 6 years of tree growth.” is not clear to me. I think it will be usefull to prepare the scheme  showing exactly the way of taking cores.

Line 230 – how many sediment cores were taken in each location?

Line 243 – it should be reference here about this usefulness of S. testudineus as a Hg bioindicator

Line 258 – after “namely” should be rather „:”, not coma

Line 308 – unnecessary word „This”

Line 307 – it should be ‘Results and discussion”

Line 308 – 332 – it should be rather in „Introduction” section

Figure 2 – please state in the Figure caption what this vertical bars mean. Is it SE or SD or something else?

In Material and methods you wrote that you took tree cores, each about 10 cm long, so why in Figure 2 you show only the cores 0 – 5 cm?

Lines 342 – 344 – „The highest levels of Hg were observed mainly in the upper layers of the Ceará A station, where the concentration of Hg was 10x higher, but decreasing to levels similar to the reference site (after 30 cm)”  - but it is not shown – I mean the data after 30 cm, and also 10x 10= 100 and for Cerea site A maximum Hg concentration is about 80 at 15 cm depth, not 100.

In general: the description of the results not correspond accuratelly with the data in the Figure 2.

Why for site A are only 5 sediment cores (0, 5, 10, 15 and 20) while for site B is 10 and 9 for site C?

Line 341 – from the figure 2 it seems to be less than 20 ng g-1. Could you be more accurate?

Line 370 – it should be  Figure 4a

Line 382 – shouldn’t be coarse sediment? And is it exactly the same as fine sediments?

Line 453 - Where are these Figures 4 A and 4B?

Line 453 – 456 – Could you provide detailed data in the form of a table or figure – not only description? Were the catched fish correlated with the sites A, B and C?

 Line 469 – there isn’t Figure 5 – only caption – so it is impossible to check the description.

Discussion should be much more results oriented. In this form it is too general.

Also conclusions from the results obtained must be provided.

Author Response

Fernando Morgado

CESAM, Department of Biology

University of Aveiro,

3810-193 Aveiro,

Portugal

30th July 2021

Dear Editor,

The authors would like to thank your considerations on the manuscript entitled “Chronological trends and mercury bioaccumulation of mercury in an aquatic semiarid ecosystem under a global climate change scenario in the North-eastern coast of Brazil”, sent for publication to Animals.

We have taken in consideration all the comments and suggestions from the reviewers and revised the manuscript accordingly.The article has been substantially improved. Following the reviewers' suggestions for corrections, an extensive review of the article was carried out. In addition to a spelling and grammar review, all the recommended structural changes were carried out: inclusion of a simple summary, abstract rewritten and enhanced with the inclusion of the suggested information, substantial reduction of the introduction (more or less 30%), separation of the results and discussion and the inclusion of a conclusion section. One new figure was also included in the materials and methods section in order to a better understanding of the study sampling design, and to increase the understanding of the texts. One new table was also included in the results containing the average Hg concentration (µg g-1) obtained in this study (sediments, fish tissues and Biota-Sediment Accumulation Factor determination (BASF)) and the maximum Hg concentrations established by international environmental guidelines standards to protect wildlife (EQS) and humans from health problems (numeric criteria for maximum acceptable concentration for Hg Baseline Values Standard for Soil and Sediments, marine water, Water for Protection of Aquatic Life, fish tissue, drinking water, and for human consumption. This table increases the ability to interpret the results in terms of risk assessment and negative effects on wildlife or human health. One new figure was also included in the results section, a conceptual model, in order to take a better visualization and understanding of the Hg dynamic on these aquatic system processes..This conceptual model represents the attempt to integrate data from these study and other published data to create an historical summary of present time and Climate Change scenario in order to understand processes that control the fate and transport of Hg in this region.

Thank you for the opportunity to improve the paper.

We hope that you will find the revised manuscript suitable for publication.

Below is our response to reviewers’ comments.

With my best regards,

Fernando Morgado (on behalf of all authors)

Reviewer #3

The topic is really interesting, however the manuscript needs many corrections and improvements before getting published. Some data (Figures) are missing, so it is impossible to check if the description is correct. You should carefully check all figures and adjust the text accordingly. Conclusions which answer the question in hypothesis must be provided.

Authors: Following the reviewers' suggestions for corrections, an extensive review of the article was carried out. In addition to a spelling and grammar review, all the recommended structural changes were carried out: inclusion of a simple summary, abstract rewritten and enhanced with the inclusion of the suggested information, substantial reduction of the introduction (more or less 30%), separation of the results and discussion and the inclusion of a conclusion section. One new figure was also included in the materials and methods section in order to a better understanding of the study sampling design, and to increase the understanding of the texts. One new table was also included in the results containing the average Hg concentration (µg g-1) obtained in this study (sediments, fish tissues and Biota-Sediment Accumulation Factor determination (BASF)) and the maximum Hg concentrations established by international environmental guidelines standards to protect wildlife (EQS) and humans from health problems (numeric criteria for maximum acceptable concentration for Hg Baseline Values Standard for Soil and Sediments, marine water, Water for Protection of Aquatic Life, fish tissue, drinking water, and for human consumption. This table increases the ability to interpret the results in terms of risk assessment and negative effects on wildlife or human health.  One new figure was also included in the results section, a conceptual model, in order to take a better visualization and understanding of the Hg dynamic on these aquatic system processes.. This conceptual model represents the attempt to integrate data from these study and other published data to create an historical summary of present time and Climate Change scenario in order to understand processes that control the fate and transport of Hg in this region

  1. Line 218 – different way of referencing to the literature than in the rest of the text – it should be as number not as the name of author.

Authors: The reference was corrected.

  1. Line 222 – does it mean that there were three trees in each site?

Authors:  Yes, correspond to 3 replicas, three trees in each site. The information has been added to the text of the article.

  1. Line 226- 229 The sentence “Trees were planted in the 1980 decade, meaning that the older trees have about 30 years and showing about 10cm radial growing from the tree center, thus each centimeter radial segment corresponds to approximately 6 years of tree growth.” is not clear to me. I think it will be usefull to prepare the scheme showing exactly the way of taking cores.

 Authors: One new figure was also included in the materials and methods section in order to a better understanding of the study sampling design, and to increase the understanding of the texts.Also, A more detailed information has been added to the text of the article. “Cores (three trees in each site), were taken at breast height (*1.5 m) from each tree using a stainless-steel borer (5 mm in diameter and 300 mm in length) (Figure 3). The trees are about 10 cm in radius. The cores were removed with only 5 to 6 cm in depth on the side of the trunk and not on the entire trunk (10cm), because a greater depth could cause distortion and compression of the core”..

  1. Line 230 – how many sediment cores were taken in each location?

Authors:  A more detailed information has been added to the text of the article. Cores (three trees in each site), were taken at breast height (*1.5 m) from each tree using a stainless-steel borer (5 mm in diameter and 300 mm in length) (Figure 3). The trees are about 10 cm in radius. The cores were removed with only 5 to 6 cm in depth on the side of the trunk and not on the entire trunk (10cm), because a greater depth could cause distortion and compression of the core”..

  1. Line 243 – it should be reference here about this usefulness of S. testudineus as a Hg bioindicator

Authors: A reference was included

  1. Line 258 – after “namely” should be rather „:”, not coma

Authors: The sentence was corrected

  1. Line 308 – unnecessary word „This”

Authors: The sentence was corrected

  1. Line 307 – it should be ‘Results and discussion”

Authors: The separation of the results and discussion sections and the inclusion of a conclusions section were carried out.

  1. Line 308 – 332 – it should be rather in „Introduction” section

Authors: The separation of the results and discussion sections and the inclusion of a conclusions section were carried out.

  1. Figure 2 – please state in the Figure caption what this vertical bars mean. Is it SE or SD or something else?

Authors: Corresponds to SD

  1. In Material and methods you wrote that you took tree cores, each about 10 cm long, so why in Figure 2 you show only the cores 0 – 5 cm?

Authors: A more detailed information has been added to the text of the article. Cores (three trees in each site), were taken at breast height (*1.5 m) from each tree using a stainless-steel borer (5 mm in diameter and 300 mm in length) (Figure 3). The trees are about 10 cm in radius. The cores were removed with only 5 to 6 cm in depth on the side of the trunk and not on the entire trunk (10cm), because a greater depth could cause distortion and compression of the core”..

  1. Lines 342 – 344 – „The highest levels of Hg were observed mainly in the upper layers of the Ceará A station, where the concentration of Hg was 10x higher, but decreasing to levels similar to the reference site (after 30 cm)” - but it is not shown – I mean the data after 30 cm, and also 10x 10= 100 and for Cerea site A maximum Hg concentration is about 80 at 15 cm depth, not 100.

Authors: A more detailed information has been added to the text of the article. The sediment cores were cut into 1cm thick slices; A first Hg quantification was made every 5 cm. In places where it seemed to indicate some oscillation/variation, + quantifications were made (eg 2cm and 2cm). That's why some colors have “fine” quantification and others only every 5 cm

  1. In general: the description of the results not correspond accuratelly with the data in the Figure 2.

Authors: A new text was made for the analysis and interpretation of figure 2. Also, Figure 2 has been modified in order to provide more detailed information provoding an easier interpretation and more accuratelly correspondent with the data

  1. Why for site A are only 5 sediment cores (0, 5, 10, 15 and 20) while for site B is 10 and 9 for site C?

Authors: A more detailed information has been added to the text of the article. The sediment cores were cut into 1cm thick slices; A first Hg quantification was made every 5 cm. In places where it seemed to indicate some oscillation/variation, + quantifications were made (eg 2cm and 2cm). That's why some colors have “fine” quantification and others only every 5 cm (coarse).

  1. Line 341 – from the figure 2 it seems to be less than 20 ng g-1. Could you be more accurate?

Authors: A new text was made for the analysis and interpretation of figure 2. Also, Figure 2 has been modified in order to provide more detailed information provoding an easier interpretation and more accuratelly correspondent with the data

  1. Line 370 – it should be Figure 4a

Authors:. An error was detected in the numbering of the figures. The authors corrected the numbering of the figures.

  1. Line 382 – shouldn’t be coarse sediment? And is it exactly the same as fine sediments?

Authors: Figure 2 refers to “sediment colors”, here they are “surface sediments”; meaning 0-2cm removable bed particles

  1. Line 453 - Where are these Figures 4 A and 4B?

Authors: An error was detected in the numbering of the figures. The authors corrected the numbering of the figures.

  1. Line 453 – 456 – Could you provide detailed data in the form of a table or figure – not only description? Were the catched fish correlated with the sites A, B and C?

Authors: For reasons unknown to the authors, probably due to an error outside the authors that occurred during the upload of the article, figure 5 was omitted. The missing figure has been reset. the authors also corrected the numbering of the figures.

  1. Line 469 – there isn’t Figure 5 – only caption – so it is impossible to check the description.

Authors: For reasons unknown to the authors, probably due to an error outside the authors that occurred during the upload of the article, figure 5 was omitted. The missing figure has been reset. the authors also corrected the numbering of the figures.

  1. Discussion should be much more results oriented. In this form it is too general.

Authors: A separation of the results and discussion sections was carried out. One new table was also included in the results containing the average Hg concentration (µg g-1) obtained in this study (sediments, fish tissues and Biota-Sediment Accumulation Factor determination (BASF)) and the maximum Hg concentrations established by international environmental guidelines standards to protect wildlife (EQS) and humans from health problems (numeric criteria for maximum acceptable concentration for Hg Baseline Values Standard for Soil and Sediments, marine water, Water for Protection of Aquatic Life, fish tissue, drinking water, and for human consumption. This table increases the ability to interpret the results in terms of risk assessment and negative effects on wildlife or human health. One new figure was also included in the results section, a conceptual model, in order to take a better visualization and understanding of the Hg dynamic on these aquatic system processes.. This conceptual model represents the attempt to integrate data from these study and other published data to create an historical summary of present time and Climate Change scenario in order to understand processes that control the fate and transport of Hg in this region. Also the inclusion of a much more oriented discussion section was carried out,

  1. Also conclusions from the results obtained must be provided.

Authors: A conclusions section was included

Round 2

Reviewer 2 Report

The quality of this version of manuscript is better, but I found some little problems. I commented the pdf file and attached it.

Author Response

Dear Reviewer

The authors would like to thank your considerations on the manuscript entitled “Chronological trends and mercury bioaccumulation of mercury in an aquatic semiarid ecosystem under a global climate change scenario in the North-eastern coast of Brazil”, sent for publication to Animals. We have taken in consideration all the comments and suggestions from the reviewers and revised the manuscript accordingly.

Thank you for the opportunity to improve the paper.

We hope that you will find the revised manuscript suitable for publication.

With my best regards,

Fernando Morgado (on behalf of all authors)

Reviewer 3 Report

The manuscript is satisfactory improved

Author Response

(The authors gave the same response as above.)
